# Novel Local Coding Algorithm for Finger Multimodal Feature Description and Recognition

**DOI:** 10.3390/s19092213

**Published:** 2019-05-13

**Authors:** Shuyi Li, Haigang Zhang, Yihua Shi, Jinfeng Yang

**Affiliations:** 1Tianjin Key Laboratory for Advanced Signal Processing, Civil Aviation University of China, Tianjin 300300, China; shuyili0909@163.com (S.L.); zhang_gang1989@126.com (H.Z.); 2Shenzhen Polytechnic, Shenzhen 518055, China; yhshi@szpt.edu.cn

**Keywords:** finger features, multimodal recognition, local coding, Gabor filter, LGS

## Abstract

Recently, finger-based biometrics, including fingerprint (FP), finger-vein (FV) and finger-knuckle-print (FKP) with high convenience and user friendliness, have attracted much attention for personal identification. The features expression which is insensitive to illumination and pose variation are beneficial for finger trimodal recognition performance improvement. Therefore, exploring suitable method of reliable feature description is of great significance for developing finger-based biometric recognition system. In this paper, we first propose a correction approach for dealing with the pose inconsistency among the finger trimodal images, and then introduce a novel local coding-based feature expression method to further implement feature fusion of FP, FV, and FKP traits. First, for the coding scheme a bank of oriented Gabor filters is used for direction feature enhancement in finger images. Then, a generalized symmetric local graph structure (GSLGS) is developed to fully express the position and orientation relationships among neighborhood pixels. Experimental results on our own-built finger trimodal database show that the proposed coding-based approach achieves excellent performance in improving the matching accuracy and recognition efficiency.

## 1. Introduction

With the arrival of the informational age and the rapid development of computer technology, people have higher requirements for the accuracy of biometric identification technology [1]. Compared with other common biometric traits, finger-based traits (e.g., fingerprint [2], finger-vein [3] and finger-knuckle-print [4],) have some advantages in uniqueness, anti-counterfeit, user acceptance, and high security [5,6,7,8]. However, affected by the external environment and the inherent differences of individuals, only relying on finger unimodal biometrics for identity authentication still has many security risks, which can no longer achieve the high-performance requirements of a user. Hence, fusing three traits from a finger together should be beneficial to address the person recognition problem [9,10]. 

However, the quality of three modal finger images is usually degraded seriously due to illumination variation in skin surfaces, which is unhelpful for reliable feature representation [11,12,13]. In addition, the finger trimodal images vary with the finger in pose rotation during imaging, which reduces the discriminability of images and further decreases the recognition accuracy rate. Therefore, exploring a robust feature representation method is very favorable for finger-based recognition improvement. 

Recently, some researchers have developed some coding-based feature expression methods, which were often considered to be able to solve the above two problems [14,15,16,17,18,19,20,21]. Ojala et al. first proposed the classical local binary pattern (LBP) algorithm for facial recognition, which has great rotation invariant and was insensitive to illumination variation [16]. In 2011, Rosdi et al. proposed a local line binary pattern (LLBP) algorithm to effectively make use of the position relationships among surrounding pixels in horizontal and vertical orientations [17]. Meng et al. [19] proposed a local direction coding (LDC) algorithm, which utilized the gradient relationships to express a venous feature for finger-vein recognition. In 2013, Peng et al. [20] combined the Gabor wavelet and LBP (GLBP) for feature extraction, which could effectively improve the ability of local and global features representation.

Noteworthy, some methods related to local graph have been presented in succession, and their variants have been successfully applied to many biometric fields [22,23,24,25,26,27,28,29]. Abusham et al. first proposed a local graph structure (LGS) algorithm to extract face features, which was insensitive to illumination [22]. However, the structure was non-symmetric, which led to a feature representation with no-equilibrium in the left and right neighborhoods. In order to balance feature representation of neighbor pixels on both sides, Mohd et al. [23] improved the original LGS operator to the SLGS operator by building a symmetrical structure. In 2015, Dong et al. [24] presented a MOW-SLGS operator for the representation of vein networks, and used the ELM to accomplish finger-vein image classification. On the basis of this information, in 2018, Yang et al. [25] put forward the Weber SLGS, which integrated differential direction features by Weber Law and the local graph structure algorithm. However, these algorithms still have some limitations in the application of representing finger multimodal features. On the one hand, the methods above only describe the relationships between the target pixel and its adjacent ones in a fixed neighborhood, while neglecting the hidden relationships among surrounding pixels. On the other hand, the assignment of different weights for symmetric pixels on left and right sides usually results in an imbalance of the feature expression in images. 

To effectively overcome these limitations, we propose a Gabor generalized symmetric local graph structure (Gabor-GSLGS) for finger multimodal fusion recognition, as shown in Figure 1. In the image capture part, first, a finger imaging device is designed, and a pose correction algorithm is proposed to reduce pose variations. A robust finger region of interest (ROI) localization approach is then employed. Secondly, a bank of 6-orientation and single-scale Gabor filters are utilized for finger ROI image enhancement. Thirdly, based on the proposed GSLGS operator, a local coding algorithm is developed for finger features representation. The coded trimodal feature images of a finger are then divided evenly into non-overlapping blocks. Thus, for a finger, we can obtain a feature vector by concatenating the histograms of all blocks. Finally, by computing the similarities between the obtained vectors, the matching results can be reported statistically. Experimental results on our established database demonstrated that the proposed feature description approach exhibits better effects than other traditional approaches in finger multimodal fusion recognition.

The reminder of this paper is organized as follows: The finger trimodal imaging device and the proposed posture correction are introduced in Section 2. The enhancement methods used for finger ROI images are described in Section 3. Section 4 details the structure of the proposed local coding algorithm. A feature matching scheme is employed to implement finger multimodal fusion recognition and described in Section 5. Section 6 outlines the extensive experiments conducted and presents the analysis of experimental results in details. Finally, Section 7 presents the summarization.

## 2. Finger Image Capture and Preprocessing 

### 2.1. Image Acquisition

As shown in Figure 2a, we have developed a homemade image acquisition device to obtain finger trimodal images. The finger imaging device is designed to capture fingerprint (FP), finger-vein (FV), and finger-knuckle-print (FKP) images automatically. It is composed of a binocular camera with two optical filters, a fingerprint acquisition instrument and an array of LEDs at a wavelength of 850 nm. In the imaging device, the FP images are directly obtained through a fingerprint instrument, which has a quick collection speed. Based on the imaging characteristics of a finger, the FV images are collected by using the near infrared (NIR) light to illuminate the palm side of a finger in penetration manner [27]. For FKP modality, we use the principle of reflecting the visible lights source for image acquisition. 

For the sake of improving convenient acquisition of images, a collection groove with fixed sizes is designed in the imaging device, which is used to limit the position of a finger for imaging. It can effectively avoid, to a large extent, an image mismatch problem caused by the rotation and translation of a finger. As shown in Figure 2b, the dimensions of our finger imaging device are 10.9 × 9.8 × 17.8 cm (length × width × height: cm).

As shown in Figure 2c, the acquisition program runs on the platform of Windows, and the software interface is built using C++ language. The top of the interface is designed to display the finger trimodal images captured in real time. Considering the friendliness of human-computer interaction, a window on the right side of the system is designed to remind users of the problems in the system operation. 

From Figure 2c, it can be clearly seen that the original captured finger trimodal images have, to a small extent, still some posture variation. To solve this problem during the acquisition process, we present the posture correction method for finger trimodal images. 

### 2.2. Posture Correction

Although a collection groove is designed in the acquisition device to fix the position of a finger, the finger still has a plane rotation at a small range. As the finger plane rotates, the edge line of the finger changes steadily. Hence, the rotation angle of a finger posture can be calculated and corrected based on the edge line of the finger. Due to the different illuminations, the edge line of the finger-vein is easier to detect and process than the finger-knuckle-print. Therefore, the finger in the finger-vein imaging space is selected to calculate the rotation angle, and then the three modalities are rotated and corrected together. The calculation process of a finger posture angle is shown in Figure 3.

At first, the captured finger image is filtered to remove noise, and the edge line of the finger is detected. Then, the point coordinates of two edge lines are extracted to calculate the midpoint coordinates. As shown in Figure 3b, {*L_n_*} (*n* = 1,2, …, *N*) represents the coordinate set in the left edge line of the finger, and {*R_n_*} represents the coordinate set in the right edge line of the finger, *X* and *Y* represent the row and column coordinates of the midpoint {*M_n_*}. The calculation is as follows:(1)$$\left\{\begin{array}{l} X_{M_{n}}=(X_{L_{n}}+X_{R_{n}})/2\\ Y_{M_{n}}=Y_{L_{n}}=Y_{R_{n}} \end{array}\right.$$

The linear fitting of the midpoint {*M_n_*} by least squares method yields the direction line: *l = kx + b*, where:(2)$$\left\{\begin{array}{l} k=\frac{\sum_{n=1}^{N}\left(x_{M_{n}}-\bar{x})(y_{M_{n}}-\bar{y}\right)}{\sum_{n=1}^{N}{\left(x_{M_{n}}-\bar{x}\right)}^{2}}\\ b=\bar{y}-k\bar{x} \end{array}\right.$$

Finally, according to *k*, the posture angle *θ* of the finger is calculated as follows:(3)$$\theta =\arctan(\frac{1}{k})$$

Noteworthy, the center of rotation should select the center of the finger direction line *l*, which can reduce the amplitude of the posture swing of the finger and improve the stability of the correction. Hence, taking the midpoint *M_N/2_* as the center of rotation and *θ* as the angle of rotation, the finger in the finger-vein imaging space is rotated and corrected. Some original images and corrected images for the same finger are shown in Figure 4.

From Figure 4, we can see that the proposed posture correction algorithm is effective to solve the problem of random plane rotation of the finger. This shows that the selection of the hardware and the posture correction algorithms have achieved better effects in improving the consistency of the finger posture.

However, from the corrected finger images, we can see that they still contain some unnecessary backgrounds and uninformative parts. Hence, the captured finger images need to be processed to implement the regions of interest (ROIs) localization.

### 2.3. ROI Extraction

Since the imaging principle and acquisition approach of FP, FV, and FKP traits are different, diverse ROI extraction methods are supposed to be adopted accordingly [2]. In this paper, we apply the core point detection method to extract the FP ROI image [30], the convex direction coding method to extract the FKP ROI image [31], and the interphalangeal joint prior method to extract the FV ROI image [3]. Therefore, the FP, FV, and FKP images are cropped into 152 × 152 pixels, 200 × 91 pixels and 200 × 90 pixels, respectively. Some finger trimodal ROI images are shown in Figure 5.

## 3. Finger Image Enhancement

In recent decades, Gabor filters have been widely applied in many fields since they not only extract the texture information in multiple directions of an image, but also reduce the influence of illumination to some extent [32]. In terms of the abundant texture information of FP, FV, and FKP traits, with respect to direction, oriented Gabor filters are used here to perform image enhancement.

A Gabor filter consists of a real part and an imaginary part. Generally, the real part is also called an even-symmetrical Gabor filter, which is suitable for ridge detection in an image [2]. Since these three modality images of a finger all have their own particular ridge textures, the real part of Gabor filter can be used to extract the flexible feature information effectively [33]. It can be expressed as
(4)$$G\left(x,y,\theta_{k},f_{k}\right)=\frac{\gamma }{2\pi \sigma^{2}}\exp\left\{-\frac{1}{2}\left(\frac{x_{\theta_{k}}^{2}+\gamma^{2}y_{\theta_{k}}^{2}}{\sigma^{2}}\right)\right\}\cos\left(2\pi f_{k}x_{\theta_{k}}\right)$$
where $x_{\theta_{k}}$ = *x*cos*θ_k_* + *y*sin*θ_k_*, $y_{\theta_{k}}$ = *y*cos*θ_k_* − *x*sin*θ_k_*, *σ* and *k* = (1, 2, …, *k*), respectively, represent the scale index and the orientation index, *θ_k_* denotes the orientation of the *k*-th Gabor filter, and *f_k_* is the central frequency of the sinusoidal plane wave. Assuming *R*(*x*,*y*) is an original ROI image, each *k*-th Gabor filtered image *I_k_*(*x*,*y*) can be obtained by
(5)$$I_{k}\left(x,y\right)=R(x,y)*G\left(x,y,\theta_{k},f_{k}\right)$$
where the symbol “∗” represents two-dimensional convolution.

First, the original ROI image is convoluted with *k*-channel Gabor filters. Then, the *k* Gabor filtered images are merged into an image *I*(*x*,*y*) by using the competitive coding method proposed in [15]. Some Gabor filtered images are shown in Figure 6. It can be clearly seen that the texture information of finger images can be effectively enhanced after multichannel Gabor filtering. Based on this, we apply the coding-based theory to obtain more stable and reliable finger features.

## 4. Feature Extraction Based on Local Coding Algorithm

To make full use of local position and gradient features between adjacent pixels in Gabor filtered images, a local coding algorithm based on generalized symmetric LGS is proposed for feature representation. The specific steps are as follows:

**Step 1** The finger trimodal images are respectively enhanced by *k*-channel even symmetric Gabor filters in Section 3, and the Gabor filtered images are obtained.

**Step 2** As shown in Figure 7, for each center pixel in the Gabor enhanced images, we respectively select three pixels in the left and right of *n × n* neighborhoods (a square area in Figure 7) to constitute the GSLGS operator in the horizontal orientation. In terms of weight distribution, the weights of symmetric pixels in the right and left sides maintain equal weights. More details are shown in Figure 7. 

Since Gabor features of finger trimodal have diverse directions, the information of the surrounding pixels in multiple orientations should be extracted for efficient feature representation. Centered on the target pixel, rotating the GSLGS operator counterclockwise by *θ_k_* (corresponding to Step 1), the structure of GSLGS in an arbitrary orientation can be obtained. For instance, when *k* = 4, the structures of GSLGS are shown in Figure 8.

**Step 3** The coding process of the GSLGS operation is shown in Figure 9. In the neighborhood of left and right sides, these gray values of the pixels are respectively compared in succession starting from each target pixel. If the value becomes larger, the relationship between the two pixels to be compared is coded to 1. In contrast, the relationship is coded to 0. The coding calculation process is expressed as
(6)$$F\left(\theta_{k}\right)=\sum_{r=1}^{6}p_{r}\left(g_{r}-f_{r}\right)2^{6-r}+\sum_{l=1}^{6}q\left(g_{l}-f_{l}\right)2^{6-l},\;\left(k=1,2,\cdots ,K\right)$$
(7)$$p_{r}\left(g_{r}-f_{r}\right)=\left\{\begin{array}{l} 1,\;g_{r}-f_{r}\geq 0,\\ 0,\;g_{r}-f_{r}<0 \end{array}\right.$$
(8)$$q_{l}\left(g_{l}-f_{l}\right)=\left\{\begin{array}{l} 1,\;g_{l}-f_{l}\geq 0,\\ 0,\;g_{l}-f_{l}<0 \end{array}\right.$$
where *g_r_* and *f_r_* (*g_l_* and *f_l_*), respectively, denote values of two adjacent pixels in the right (left) neighborhood, and *F(θ_k_)* represent the feature coded value in the *θ_k_* orientation. 

From Figure 9, we can see that the coded values of the target pixel at 0° and 45°, respectively, can be obtained according to Equations (6)–(8). Similarly, the same calculation process is done at 90° and 135°. Thus, we calculate the coded values of the central pixel in these four directions as follow:
*F*(*θ*_1_) = (010100)_2_ + (110110)_2_ = (0 × 32 + 1 × 16 + 0 × 8 + 1 × 4 + 0 × 2 + 0 × 1) + (1 × 32 + 1 × 16 + 0 × 8 + 1 × 4 + 1 × 2 + 0 × 1) = 74.*F*(*θ*_2_) = (100100)_2_ + (000100)_2_ = (1 × 32 + 0 × 16 + 0 × 8 + 1 × 4 + 0 × 2 + 0 × 1) + (0 × 32 + 0 × 16 + 0 × 8 + 1 × 4 + 0 × 2 + 0 × 1) = 40.*F*(*θ*_3_) = (100110)_2_ + (010110)_2_ = (1 × 32 + 0 × 16 + 0 × 8 + 1 × 4 + 1 × 2 + 0 × 1) + (0 × 32 + 1 × 16 + 0 × 8 + 1 × 4 + 1 × 2 + 0 × 1) = 60.*F*(*θ*_4_) = (010011)_2_ + (010101)_2_ = (0 × 32 + 1 × 16 + 0 × 8 + 0 × 4 + 1 × 2 + 1 × 1) + (0 × 32 + 1 × 16 + 0 × 8 + 1 × 4 + 0 × 2 + 1 × 1) = 39.

**Step 4** Inspired by the optimal response of Gabor filters in multiple orientations, we choose the maximum value among these coded values as the final coded value, which can be defined as
(9)$$F\left(x,y\right)=\arg\underset{\theta_{k}\in (0^{{}^{\circ}},180^{{}^{\circ}})}{\max}\left\{F\left(\theta_{k}\right)\right\}$$

As mentioned above, the final coded value of each target pixel in the Gabor enhanced image can be obtained according to the GSLGS operator. For instance, the coded value in Figure 7 is: *F*(*x*,*y*) = argmax {*F*(*θ_1_*), *F*(*θ_2_*), *F*(*θ_3_*), *F*(*θ_4_*)} = *F*(*θ_1_*) = 74.

Considering the great capability of a Gabor filter in enhancing texture feature from any orientation, the GSLGS operator is extended into arbitrary orientations, which has superior orientation selectivity. Therefore, it can effectively solve image mismatch problem due to finger pose variation. More importantly, the proposed local coding algorithm can entirely consider the relationships between each target pixel and its surrounding neighborhoods. In addition, the distribution of weights is conformable in the symmetric pixels on both sides. Hence, the finger feature representation of local neighborhoods can maintain balance in the GSLGS operator.

## 5. Feature Fusion and Matching

In this section, a gray histogram-based feature matching method is used for finger trimodal fusion recognition, as shown in Figure 10. First, the coded finger trimodal images are uniformly separated into *M* non-overlapping division blocks. Then, the *M* local histograms corresponding to each sub-block are established, respectively. Assuming that *H_fv_^i^*(*I* = 1, 2, …, *M*) represents the histogram of the *i*th division block in a coded finger-vein image, the global histogram *H_fv_* is defined as
(10)$$H_{fv}=\left(H^{1},H^{2},\cdots ,H^{M}\right)$$

Similarly, *H_fp_* and *H_fkp_* represent the global histogram of a coded fingerprint image and finger-knuckle-print image. Then, the final feature histogram *H* of a finger trimodal image can be expressed by
(11)$$H=\left(H_{fp},H_{fv},H_{fkp}\right)$$

After the above calculation, we can obtain the feature histogram of each finger sample. Here, we can use various classification algorithms, such as SVM, ELM and *k*-NN [34]. In this section, for convenience, the intersection coefficient between two feature vectors is calculated to determine the similarity of two individuals [29]. Assuming *H_1_(i)* and *H_2_(i)* denote the histograms of two samples to be matched, the similarity can be computed by
(12)$$sim\left(H_{1}(i),H_{2}(i)\right)=\frac{\sum_{i=1}^{L}\min[H_{1}(i),H_{2}(i)]}{\sum_{i=1}^{L}H_{1}(i)}$$
where *L* denotes the dimension of a feature vector to be matched. In the matching process, if the intersection coefficient *sim*(·) is >T (similarity decision threshold), it means that the two samples are similar and are able to be matched. But if the intersection coefficient *sim*(·) is ≤T, it means that the two samples are not matched. Thus, two samples will tend to be more similar as the intersection coefficient increases. The similarity decision threshold T corresponds to the threshold value when the false rejection rate (FRR) is the same as the false accept rate (FAR).

## 6. Experimental Results

In order to verify the proposed coding-based method, a finger trimodal database from a homemade image acquisition system is used in our experiments. The database contains a total of 17,550 images from 585 individual fingers (index finger, middle finger, and ring finger) of both hands, and each finger contains 30 images (10 images per modality). Here, we randomly select 3000 images samples from 100 different individuals, each of which, respectively, contains 10 images on the FP, FV, and FKP traits, as the experimental database.

Here, the proposed Gabor-GSLGS algorithm is implemented using MATLAB R2014a on a standard desktop PC which is equipped with Inter Core i5-7400 CPU 3 GHz and 8 GB RAM. 

The detailed experiments are as follows: In Section 6.1, we mainly describe the analysis of the influence of different parameter selection on the recognition rate. Section 6.2 presents the detailed comparison of the performance of unimodal and multimodal recognition. The experimental results of different feature extraction methods are compared in Section 6.3.

### 6.1. Parameter Selection

#### 6.1.1. Selection of *k*

On the basis of the above introduction in Section 3 and Section 4, we can see that the number of orientations in the local coding algorithm corresponds to the number of channels in the Gabor filter. Hence, different *k* values produce different effects on the performance of the finger multimodal recognition. In order to find the optimal parameters of *k*, we evaluate it using two recognition indicators, equal error rate (EER) and the time cost of feature extraction. EER listed in Table 1 is the error rate where FRR and FAR are equal. Here, FAR indicates the identification result of incorrect acceptation for an individual, while FRR demonstrates the result of incorrect rejection. The ROC (receiver operating characteristic) curves for intersection coefficient measures are plotted in Figure 11, where FAR and FRR are shown in the same plot at different thresholds.

From Figure 11, we can see that the EER is lowest when *k* is 6. However, as the value of *k* increases, the time cost of finger feature extraction also increases. Considering recognition efficiency and accuracy, the parameter *k* corresponding to 6 is selected in following experiments.

#### 6.1.2. Selection of Neighborhood and Image Division

Apart from parameters *k*, the size of the neighborhood *n × n* that constitutes the structure of the GSLGS operator and the number of image division blocks *M* are also critical factors for finger trimodal recognition. Considering that *n* and *M* have a great influence on the recognition performance of the proposed algorithm, therefore, it is important to select suitable parameters. Here, we select different neighborhoods and image block sizes to perform the experiments. Some EERs of different parameters are listed in Table 2, with some ROCs shown in Figure 12 and Figure 13.

From Figure 12, it can be clearly seen that the ROC curves vary by changing *n* (*n* = 3, 5 or 7, respectively). This shows that different neighborhoods have different effects on the performance of finger trimodal recognition. By observing these obtained curves in the condition of the same image division block, such as *M* = 6, we find that the EER is lowest when the size of the neighborhood is selected as 5 × 5 (*n* = 5). Similarly, when *M* = 7, 8 or 9, respectively, the pixels selected in a 5 × 5 neighborhood for constructing the GSLGS operator also have optimal accuracy. The reason is that a 3 × 3 neighborhood is more sensitive to noise, while the 7 × 7 neighborhood is relatively weak in the capability of feature expression. However, the 5 × 5 neighborhood is preferred for feature expression among surrounding pixels and is insensitive to noise. Hence, *n* = 5 is the optimal parameter for constructing the proposed GSLGS operator. 

The experimental results of different division blocks by using GSLGS with a 5 × 5 neighborhood are shown in Figure 13. From Figure 13, we can find that the proposed local coding algorithm obtains the best accuracy when the number of division blocks is 8 × 8 (*M* = 8). This shows that an appropriate image division scheme is beneficial for improving recognition accuracy rate. Hence, the image division blocks *M* = 8 is an optimal choice for the proposed Gabor-GSLGS approach in finger trimodal recognition. 

### 6.2. Comparison of Unimodal and Multimodal

The proposed local coding algorithm of finger trimodal can also be applied for finger single modal recognition. Here, the experiments of finger unimodal and multimodal recognition are performed when *n* = 5 and *M* = 8. The experimental results of different modal combinations are listed in Table 3.

From Figure 14, we can see that the EER rate of different modal combinations are different. It is noted that the bimodal combination (FV + FKP and FV + FP) can achieve a better accuracy than single modal, especially for the FP trait and FKP trait. 

From Table 3, we can find that three modal combination have the best recognition accuracy, while the time cost increases with the increase of the modality number. It is noteworthy that in single modal recognition, FV trait performs better than FP trait and FKP trait. This shows that the FV trait is the most dominant trait in the three modalities.

In total, these results show that multimodal fusion recognition performs better than single modal. The reason is that the multimodal combination can make full use of the discrimination of different modalities and different modalities can complement each other in multimodal fusion recognition. However, the computational efficiency of multimodal recognition can still be improved. 

### 6.3. Comparison of Different Methods

In order to further evaluate the proposed local coding method, here we compare it with some common feature extraction methods (LBP [16], GLBP [20], LLBP [17], SLGS [23], and MOW-LGS [24]). The ROCs are plotted in Figure 15, and the simulation results are listed in Table 4. 

From Figure 15, we observe that the EER of the proposed approach is lowest among these feature extraction methods. Hence, feature expression based on the proposed coding approach can effectively address the problem of illumination and finger posture variation in finger trimodal fusion recognition. 

From Table 4, we can clearly see that the proposed local coding algorithm not only produces the best recognition accuracy, and also that the computation cost of feature extraction is also lowest as compared with other methods. This shows that our method is more robust to finger feature representation.

## 7. Conclusions

In this paper, a posture correction approach was first designed for reducing the finger pose variation. To solve the problem that the feature expression method was sensitive to illumination variation and posture rotation, a novel local coding algorithm was proposed for finger trimodal fusion recognition. On the one hand, the Gabor filter, to some extent, can effectively reduce the influence of illumination and noise in an image. On the other hand, the posture correction method and the local coding method were used to address the problem of finger posture variation. The proposed Gabor-GSLGS algorithm made full use of the texture features in multiple orientations between surrounding pixels. Furthermore, the proposed method assigned the same weights in symmetrical pixels, which improved the equilibrium of the feature representation of the finger images. The experimental results showed that our method could improve the accuracy and computational efficiency of finger trimodal fusion recognition. 

As part of our future work, we will apply the proposed local coding algorithm to other public biometric databases. Moreover, we will focus on reducing the dimensions of the feature vector and improving the efficiency of finger multimodal fusion recognition. At the same time, we will aim to exploit a more robust and effective fusion method which can integrate multiple modal features for personal identification. 

## Figures and Tables

**Figure 1 sensors-19-02213-f001:**
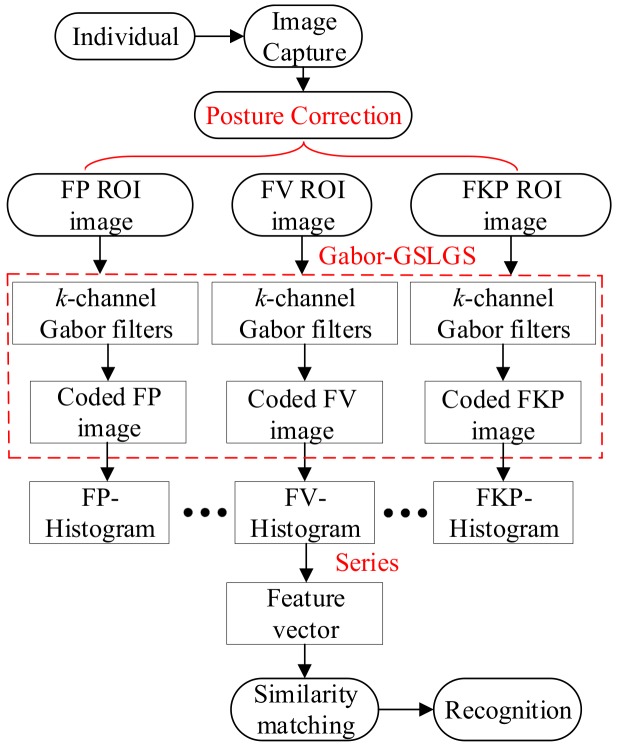
Finger multimodal recognition process based on the Gabor generalized symmetric local graph structure (Gabor-GSLGS).

**Figure 2 sensors-19-02213-f002:**
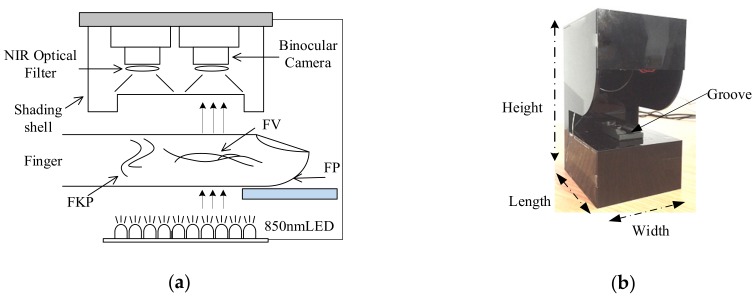
A finger trimodal image acquisition system. (**a**) the imaging schematic diagram; (**b**) a homemade image capture device; (**c**) a system interface of image acquisition.

**Figure 3 sensors-19-02213-f003:**
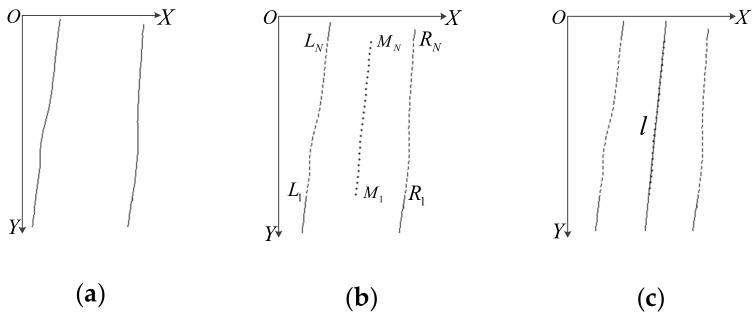
Computing finger posture angle. (**a**) the edge line of the finger; (**b**) the coordinate extraction of the finger edge line; (**c**) finger rotation direction extraction.

**Figure 4 sensors-19-02213-f004:**
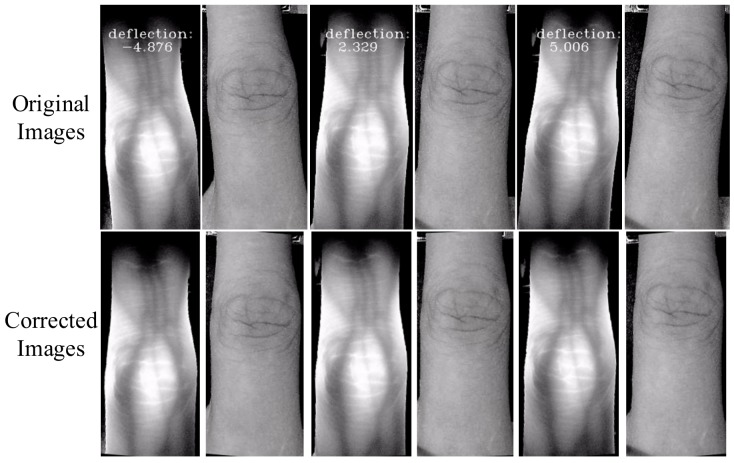
Some corrected image samples after rotation.

**Figure 5 sensors-19-02213-f005:**
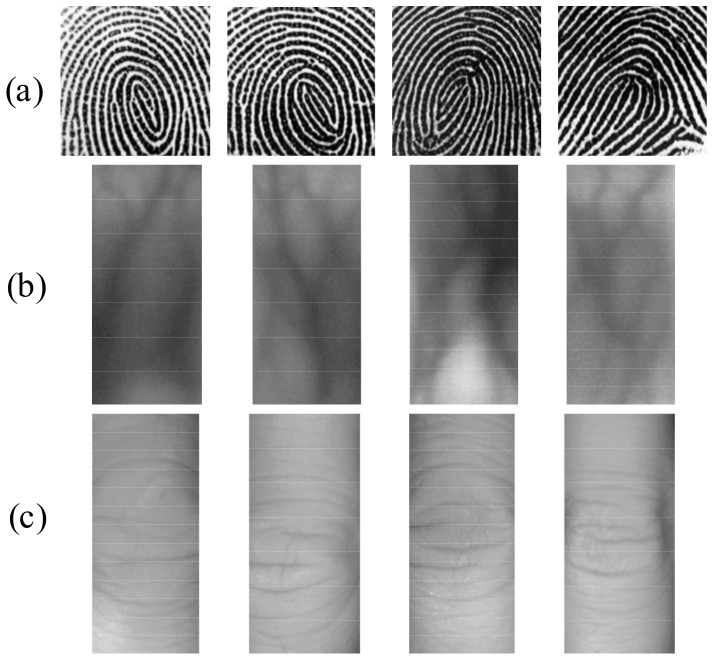
The finger trimodal region of interest (ROI) images of four fingers. (**a**) fingerprint (FP) ROIs samples; (**b**) samples of finger-vein (FV) ROIs; (**c**) finger-knuckle-print (FKP) ROIs samples.

**Figure 6 sensors-19-02213-f006:**
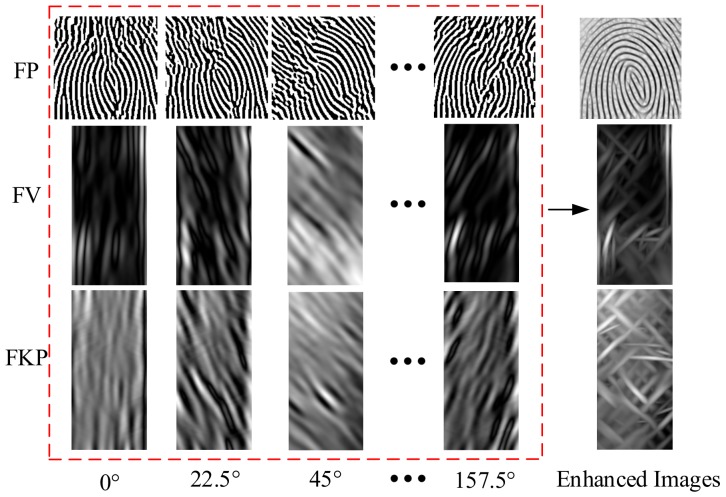
The enhanced images of the finger three modalities.

**Figure 7 sensors-19-02213-f007:**
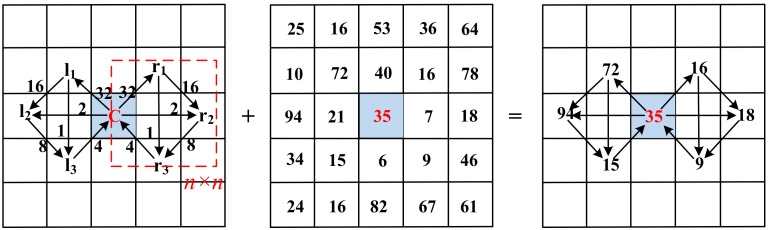
The design of the GSLGS operator (0° direction, *n* = 3).

**Figure 8 sensors-19-02213-f008:**
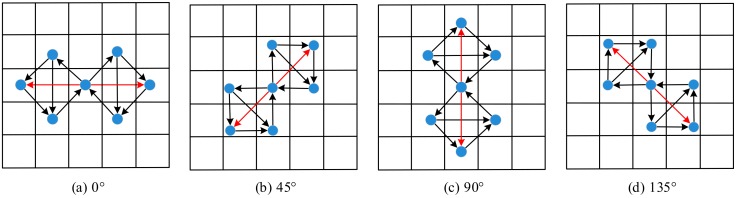
The GSLGS operator (*k* = 4).

**Figure 9 sensors-19-02213-f009:**
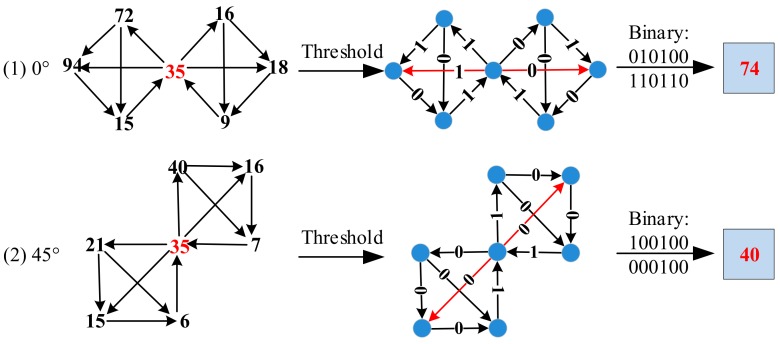
The coding process of GSLGS operator at 0° and 45°.

**Figure 10 sensors-19-02213-f010:**
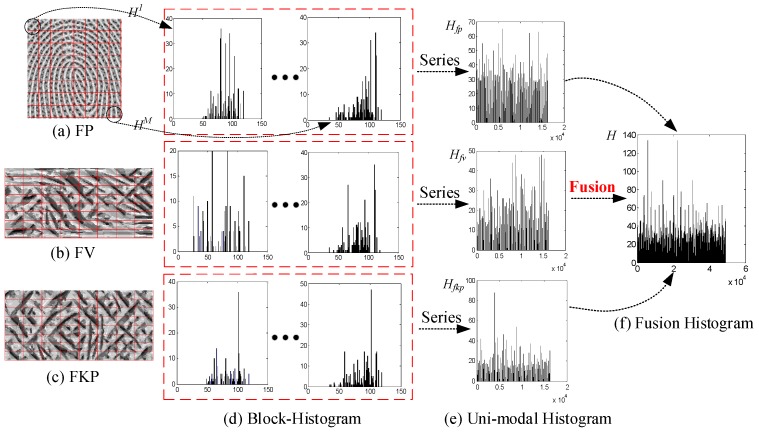
The fusion of finger trimodal features.

**Figure 11 sensors-19-02213-f011:**
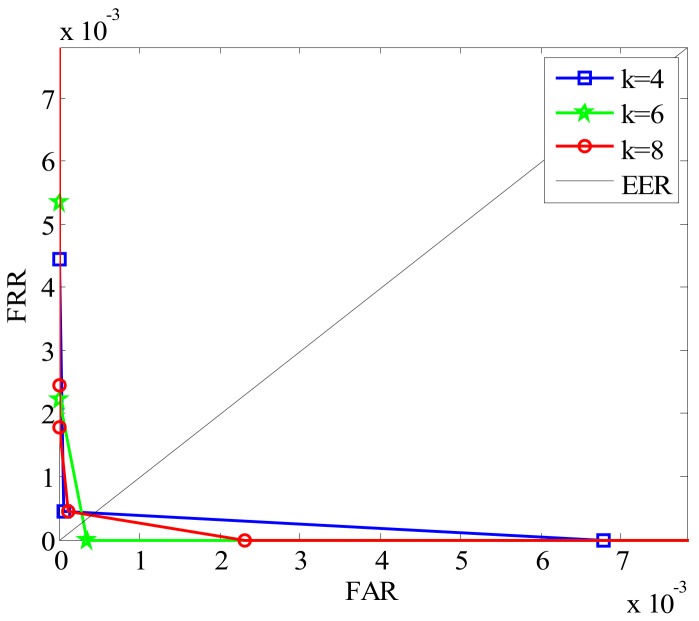
Receiver operating characteristic (ROC) of different *k*.

**Figure 12 sensors-19-02213-f012:**
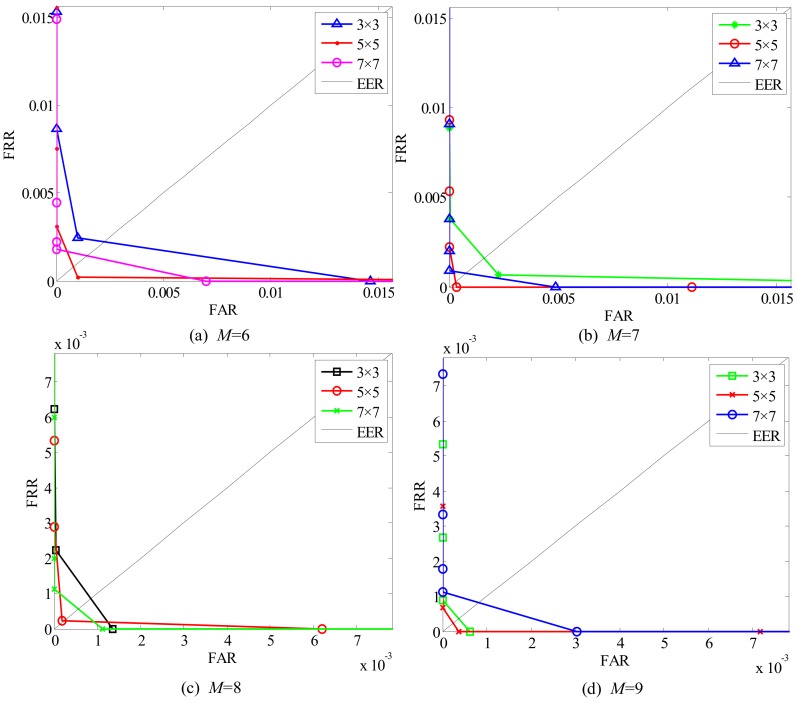
ROC of different neighborhoods in *M* = 6, 7, 8, 9.

**Figure 13 sensors-19-02213-f013:**
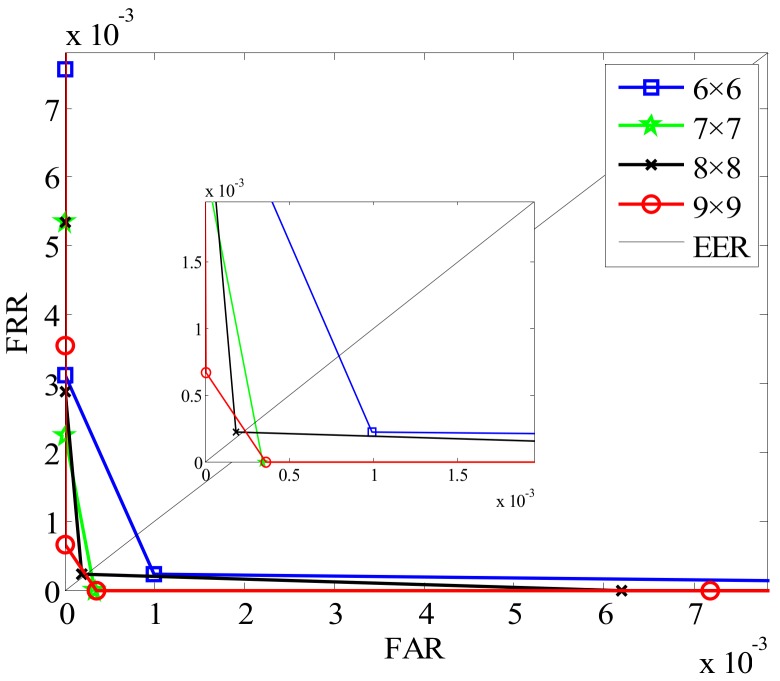
ROC of different division blocks *M* in a 5 × 5 neighborhood.

**Figure 14 sensors-19-02213-f014:**
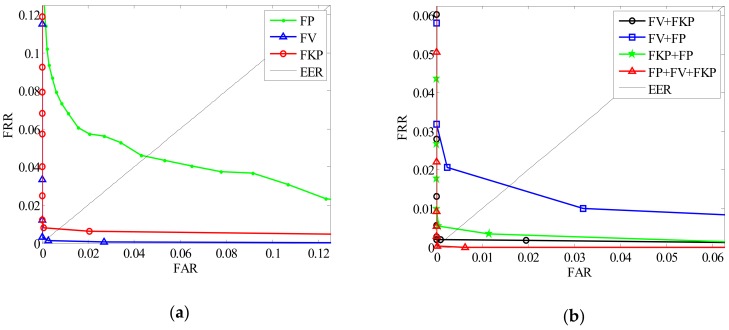
Comparison results of different modal combinations. (**a**) ROC of unimodal recognition; (**b**) ROC of multimodal recognition.

**Figure 15 sensors-19-02213-f015:**
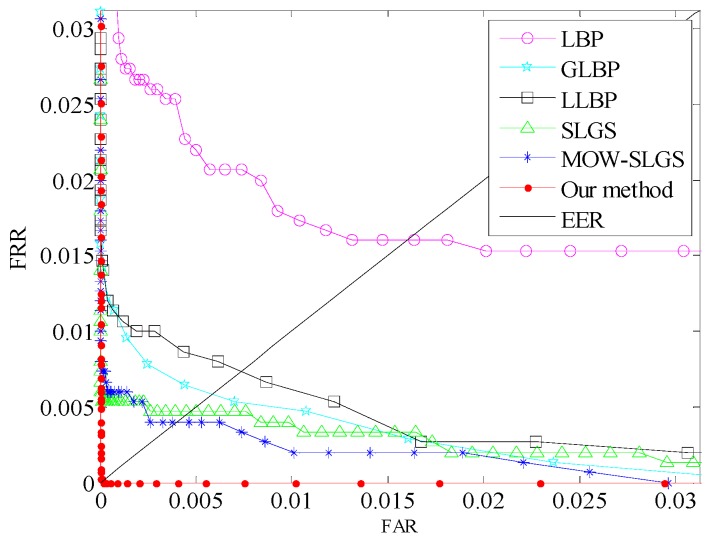
Comparisons of different methods.

**Table 1 sensors-19-02213-t001:** Comparisons on equal error rate (EER) (%) and time cost (single individual).

*k*	4	6	8
EER (%)	0.042	0.029	0.038
Time cost (s)	0.012	0.017	0.031

**Table 2 sensors-19-02213-t002:** Comparisons on EER(%) for different parameters.

	Blocks	6 × 6	7 × 7	8 × 8	9 × 9
Neighborhood					
**3 × 3**		0.22	0.16	0.086	0.37
**5 × 5**		0.08	0.029	0.022	0.024
**7 × 7**		0.14	0.075	0.056	0.082

**Table 3 sensors-19-02213-t003:** Comparisons on EER (%) and time cost (single individual).

Modal	FP	FV	FKP	FV + FKP	FV + FP	FKP + FP	FP + FV + FKP
EER (%)	4.26	0.19	0.40	0.20	0.16	0.46	0.022
Time cost (s)	0.015	0.010	0.015	0.021	0.019	0.018	0.029

**Table 4 sensors-19-02213-t004:** Comparisons on EER (%) and time cost (single individual).

Methods	LBP	GLBP	LLBP	SLGS	MOW-LGS	Our Method
EER (%)	1.60	0.58	0.74	0.46	0.42	0.022
Time cost (s)	0.129	0.186	0.261	0.110	0.192	0.029

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
