# Peer review of "Novel Local Coding Algorithm for Finger Multimodal Feature Description and Recognition"

_sensors, 2019, doi:10.3390/s19092213_

Round 1

Reviewer 1 Report

The authors need to clarify the followings in order to improve the manuscript:

- Does the system support both hands, or just left or right hand only? And what digit (finger)?

- Details of the demography of the image dataset?

- Explain more about the evaluation metrics.

- Details of the experiments. A flowchart should be included. Were all the images tested?

- The authors should compare the proposed method with using the centroid displacement-based distance proposed in 10.1109/THMS.2015.2453203 (DOI) (similar to using an enhanced 1-NN algorithm).

Author Response

Response to Reviewer 1 Comments

Point 1: Does the system support both hands, or just left or right hand only? And what digit (finger)?

Response 1: Yes, the system support both left and right hands. During the registration process, the system can capture the fingers of both hands. During the identification process, the registered finger is used for identification. Our homemade finger trimodal database contains multiple fingers of the left and right hands, and the finger refers to index finger, middle finger and ring finger. More detail can be referred in the first paragraph of Section 6.

Point 2: Details of the demography of the image dataset?

Response 2: In the modified paper, we have made a detailed description for the demography of the image database. Please see the Section 6 on Page 10. The details are as follows: The database totally contains 17550 images from 585 individual fingers (index finger, middle finger, and ring finger) of both hands, and each finger contains 30 images (10 images per modality).

Point 3: Explain more about the evaluation metrics.

Response 3: Considering this suggestion, we have explained more about the evaluation metrics. Please see the first paragraph in Subsection 6.1.1. The details are as follows: EER listed in Table 1 is the error rate where FRR and FAR are equal. Here, FAR indicates the identification result of incorrectly acceptation for an individual, while FRR demonstrates the result of incorrectly rejection. The ROC (Receiver Operating Characteristic) curves for intersection coefficient measures are plotted in Figure 11, where FAR and FRR are shown in the same plot at different thresholds.

Point 4: Details of the experiments. A flowchart should be included. Were all the images tested?

Response 4: 

(1) According to the Reviewers comments, we have presented more details of the experiments in the second paragraph of Section 6. And the flowchart is shown in Figure 1. 

The specific contents are as follows: Here, the proposed Gabor-GSLGS algorithm is implemented using MATLAB R2014a on a standard desktop PC which is equipped with a Inter Core i5-7400 CPU 3 GHz and 8 GB RAM.

The detailed experiments are as follows: In Section 6.1, we mainly analyze the influence of different parameter selection on the recognition rate. Section 6.2 presents the detailed comparison of the performance of uni-modal and multi-modal recognition. The experimental results of different feature extraction methods are compared in Section 6.3.

(2) Experiments were performed on the entire database, so all the images were tested.

Point 5: The authors should compare the proposed method with using the centroid displacement-based distance proposed in 10.1109/THMS.2015.2453203 (DOI) (similar to using an enhanced 1-NN algorithm).

Response 5: Thank you very much for your comments. We have carefully read the proposed method in 10.1109/THMS.2015.2453203 (DOI), which is innovative and very effective for palm recognition. However, our paper focuses on the proposed coding-based feature extraction method and multi-modal feature fusion method, rather than conducting the comparison of the classification algorithms.

After obtaining the feature vector (histogram) of each finger sample, we can use various classification algorithms for image classification, such as SVM, ELM, k-NN and the centroid displacement-based distance algorithm. More detail can be referred in the third paragraph of Section 5. In this paper, for convenience, we simply calculate the intersection coefficient of histogram to determine the similarity of two individuals.

Therefore, in the simulation experiment, we mainly analyze the results of the proposed feature extraction method and feature fusion method, instead of verifying the performance of the classification methods. 

Special thanks to you for your good comments.

Reviewer 2 Report

The Authors propose a new algorithm for finger multi-modal feature description and recognition. The multi-modality includes three elements: fingerprint (FP), finger-vein (FV) and finger-knuckle-print (FKP). The method uses multi-orientation Gabor filters for finger images enhancement and a local coding for finger features representation. Experiments performed with a database prepared by the Authors show that the proposed approach gives better recognition efficiency than other traditional approaches.

Remarks

Please provide details enabling the reproducibility of the results of the experiments.

The descriptions in sections 4, 5 are imprecise. The reader has to guess what GSLGS looks like for n = 5, 7. Figure. 10 is not understandable.

The protocol for determining the effectiveness of recognition is not clearly stated.

line 22: database shown that

Figure 1 FP, FV, FKP Histogrom

Line 73: a multi-orientation Gabor filters are

line163: why cos/sinθ instead of cos/sinθk?

line 183: are maintain conformable

Author Response

Response to Reviewer 2 Comments

Point 1: The descriptions in sections 4, 5 are imprecise. The reader has to guess what GSLGS looks like for n = 5, 7. Figure. 10 is not understandable.

Response 1: In the modified paper, we have made detailed and precise description of the proposed Gabor-GSLGS algorithm. Some major modifications are as follows:

(1) In Section 4, we have improved the description about the constitution of the proposed GSLGS operator. Please see the Step 2 part in Section 4. The revised content is: As shown in Figure 7, for each center pixel in Gabor enhanced images, we respectively select three pixels in the left and right of n×n neighborhoods (a square area in Figure 7) to constitute the GSLGS operator in horizontal orientation.

(2) In order to better understand the fusion process of the finger trimodal features, we have made some adjustments to Figure 10, in which more detailed description about the gray histogram calculation of the finger trimodal image are presented. More detail can be referred in Figure 10 of the Section 5.

Point 2: The protocol for determining the effectiveness of recognition is not clearly stated.

(1) line 22: database shown that

(2) Figure 1: FP, FV, FKP Histogrom

(3) line 73: a multi-orientation Gabor filters are

(4) line163: why cos/sinθ instead of cos/sinθk?

(5) line183: are maintain conformable

Response 2: According to the reviewers comments, we made correction to more clearly state the protocol for determining the effectiveness of recognition. Some major modifications are as follow:

(1) In the abstract, we have modified the description about the database. The statements of database shown that were corrected as finger trimodal database show that.

(2) We are so sorry for the mistake in the Figure 1. The statements of FP, FV, FKP Histogrom were corrected as FP, FV, FKP Histogram, which means feature histogram of coded FP, FV, FKP images. More detail can be referred in Figure 1 of the introduction.

(3) Considering the reviewers suggestion, we have made a modification in order to more clearly describe the used Gabor filters. The statements of a multi-orientation Gabor filters are were corrected as a 6-orientation and single-scale Gabor filters are. Please see the fifth sentence in the introduction.

(4) We are very sorry for the mistake in the line163. We have corrected it. The statements of cos/sinθ were corrected as cos/sinθk.

We are very sorry for our incorrect writing in the line183, We have corrected it. The statements of are maintain conformable were corrected as maintain equal. Please see the Step2 part in the section 4

Round 2

Reviewer 1 Report

I am OK with the response from the authors.

Reviewer 2 Report

I accept the coorections made  based on my review.